

# Machine learning algorithm-based biomarker exploration and validation of mitochondria-related diagnostic genes in osteoarthritis

Hongbo Wang[1,2,*], Zongye Zhang[2,*], Xingbo Cheng[2], Zhenxing Hou[2], Yubo Wang[3], Zhendong Liu[2] and Yanzheng Gao[2]

[1] Department of Urology Surgery, Lanzhou University Second Hospital, Lanzhou, Gansu, China

[2] Department of Surgery of Spine and Spinal Cord, Henan Provincial People's Hospital, People's Hospital of Zhengzhou University, Zhengzhou, Henan, China

[3] School of Basic Medicine and Forensic Medicine, Henan University of Science & Technology, Luoyang, Henan, China

[*] These authors contributed equally to this work.

Corresponding authors
Yanzheng Gao, yanzheng-gaohn@163.com
Zhendong Liu, superliuyisheng@outlook.com

## ABSTRACT

The role of mitochondria in the pathogenesis of osteoarthritis (OA) is significant. In this study, we aimed to identify diagnostic signature genes associated with OA from a set of mitochondria-related genes (MRGs). First, the gene expression profiles of OA cartilage GSE114007 and GSE57218 were obtained from the Gene Expression Omnibus. And the limma method was used to detect differentially expressed genes (DEGs). Second, the biological functions of the DEGs in OA were investigated using Gene Ontology (GO) and Kyoto Encyclopedia of Genes and Genomes (KEGG) enrichment analysis. Wayne plots were employed to visualize the differentially expressed mitochondrial genes (MDEGs) in OA. Subsequently, the LASSO and SVM-RFE algorithms were employed to elucidate potential OA signature genes within the set of MDEGs. As a result, GRPEL and MTFP1 were identified as signature genes. Notably, GRPEL1 exhibited low expression levels in OA samples from both experimental and test group datasets, demonstrating high diagnostic efficacy. Furthermore, RT-qPCR analysis confirmed the reduced expression of Grpel1 in an in vitro OA model. Lastly, ssGSEA analysis revealed alterations in the infiltration abundance of several immune cells in OA cartilage tissue, which exhibited correlation with GRPEL1 expression. Altogether, this study has revealed that GRPEL1 functions as a novel and significant diagnostic indicator for OA by employing two machine learning methodologies. Furthermore, these findings provide fresh perspectives on potential targeted therapeutic interventions in the future.

## INTRODUCTION

Osteoarthritis (OA), a prevalent degenerative joint disease, significantly contributes to chronic disability among older individuals (*Hu et al., 2022*). Presently, the primary therapeutic approaches for OA encompass non-steroidal anti-inflammatory drugs and surgery interventions (*Abramoff & Caldera, 2020*). Although the symptoms of OA

can be controlled to varying degrees, these treatments still have certain limitations and risks. For example, gastrointestinal risks from long-term use of NSAIDs, surgical complications associated with surgical treatments, and problems with aging and wear of joint materials (*Curlewis et al., 2023*). Therefore, it is important for OA patients if some effective biomarkers can be developed to help diagnose and treat synovial or cartilage degeneration and achieve early intervention to avoid surgery.

Mitochondria are essential organelles within cells, regulating key functions such as energy metabolism, apoptosis, and calcium homeostasis (*Annesley & Fisher, 2019*). In patients with OA, chondrocytes often exhibit mitochondrial dysfunction, including reduced activity of electron transport chain complexes and increased oxidative stress (*Mao et al., 2020*). Specifically, the reduced activity of electron transport chain complexes I, II, and III leads to energy deficiency, affecting chondrocytes' normal function and metabolic activities (*Blanco, Rego & Ruiz-Romero, 2011*). The increase in oxidative stress also activates inflammatory signaling pathways, increasing the expression and release of cytokines such as TNF-α and IL-1β, further damaging chondrocytes and degrading the cartilage matrix (*Liao et al., 2020*). Therefore, the role of mitochondria-related genes in the pathological process of OA has garnered researchers' attention. For instance, the AMPK-SIRT3 axis regulates mitochondrial homeostasis in chondrocytes, playing a crucial role in the progression of OA (*Chen et al., 2021b*). Overexpression of BNIP3 leads to cartilage degradation and chondrocyte apoptosis by activating mitophagy, making it a potential therapeutic target for OA (*Kim, Song & Jin, 2021*). Low expression of LONP1 accelerates OA progression by inducing mitochondrial dysfunctions such as oxidative stress, metabolic changes, and mitophagy, enhances inflammatory responses, increases apoptosis, and exacerbates matrix degradation by activating the MAPK pathway (*He et al., 2022*). Additionally, FUNDC1 induces mitophagy to maintain chondrocyte homeostasis, and its knockout worsens OA progression, while PFKP interaction enhances this process (*Fang et al., 2023*). In summary, exploring the mechanisms of mitochondria-related genes in OA is significant, providing new insights for identifying potential therapeutic targets for OA.

However, when it comes to the discovery of disease biomarkers, traditional molecular biology research has the following shortcomings: Firstly, it is more difficult to analyze complex molecular regulatory networks in disease from a comprehensive perspective; Second, the small experimental sample size makes it difficult to find biomarkers with better specificity. For this reason, an increasingly large number of scholars have begun to explore disease biomarkers using bioinformatics. The feature selection methods represented by Least Absolute Shrinkage and Selection Operator (LASSO) regression analysis and the support vector machine- recursive feature elimination (SVM-RFE) algorithm can identify important disease-related signature genes from disease transcriptome data, providing valuable information for early diagnosis and treatment (*Wu et al., 2022*; *Zhang et al., 2022*). Hence, if more valuable biomarker can be extracted from mitochondria-related genes (MRGs) by LASSO regression combined with the SVM-RFE algorithm, it is of great significance for the diagnosis and treatment of OA.

In the present study, transcriptomic data from OA cartilage samples available in Gene Expression Omnibus (GEO) were used to explore potential disease signature genes
in mitochondria-related genes (MRGs). The aim was to assess the value of these genes in disease diagnosis and provide reliable targets for OA diagnosis and treatment. Additionally, this study also preliminarily investigated the relationship between the expression of the signature genes and the immune microenvironment of OA, to improve the understanding of the pathogenesis of the disease. Meanwhile, an *in vitro* model of OA was also established, and the expression of signature genes was verified using RT-qPCR to increase the reliability of the results. Consequently, the present study may improve understanding of the underlying molecular mechanisms of OA and may contribute to its diagnosis and treatment.

## METHODS

### Data collection
The study utilized data from the GEO, using GSE114007 as the experimental dataset to identify signature genes associated with OA, which included 18 normal and 20 OA cartilage samples. GSE57218 was used as the validation dataset to confirm the results of the first study, which included seven normal and 33 OA cartilage samples.

### Identification of differentially expressed genes and mitochondria-related genes
Differentially expressed genes (DEGs) were analyzed with the R package "limma". Screening criteria were $|Log2FC| >= 1$ and adjust P. Val $< 0.05$. A list of mitochondria-related genes was also obtained for subsequent studies. Specifically, the list of MRGs was obtained by searching for mitochondria in the Molecular Signatures Database (https://www.gsea-msigdb.org/gsea/msigdb/index.jsp) using the keyword 'mitochondria', and the specific information is shown in Table S1.

### Functional enrichment analysis of differentially expressed genes
To initially explore the functions and pathways played by DEGs in OA initiation and progression, enrichment analyses were performed. Gene Ontology (GO) and Kyoto Encyclopedia of Genes and Genomes (KEGG) enrichment analyses were performed using the "clusterProfiler" package. Adjust P.Val 0.05 indicates statistically significant differences. The results were visualized by using the "ggplot2" package, and the corresponding histograms and bubble plots were obtained.

### Disease ontology analysis of differentially expressed genes
Disease ontology (DO) provides a unified description of genes from the perspective of diseases, which can be used to reveal the relationship and function between genes and diseases, thereby deepening the understanding of diseases and the development of therapeutic methods. In this study, disease ontology enrichment analysis of DEGs was performed using the "DOSE" package of the R, with the threshold set at adj.P.Val $< 0.05$. The analysis results were visualised using histograms and bubble plots.

### LASSO regression analysis and SVM-RFE algorithm
To discover highly specific signature genes, this study combined two machine learning methods, LASSO and SVM-RFE, to improve the accuracy of signature gene selection.

LASSO, implemented using the "glmnet" package, employs L1 regularization to effectively identify key features from high-dimensional linear data. SVM-RFE, implemented using the "e1071", "kernlab", and "caret" packages, recursively eliminates the least important features, making it suitable for handling nonlinear and small-sample data. By combining these methods, we leveraged their complementary strengths to enhance the robustness and precision of feature selection. Subsequently, a Venn diagram was used to identify the intersecting signature genes selected by both methods, ensuring the highest discriminatory signature genes were identified.

## OA signature gene expression and diagnostic efficiency

To assess the accuracy and diagnostic value of the OA signature genes identified in Section 'Differential expression analysis of signature genes and diagnostic efficacy for diseases', we first analyzed the expression levels of the intersecting genes selected by LASSO and SVM-RFE algorithms using the "limma" and "ggpubr" packages and visualized the results using box plots. Following this, we employed the "pROC" package to generate receiver operating characteristic (ROC) curves and determine the area under the curve (AUC) values, thereby assessing the diagnostic efficacy of these signature genes for OA.

## Immune infiltration analysis

It is worth noting immune cell-mediated inflammatory responses significantly contribute to the development of OA. Single-sample gene set enrichment analysis (ssGSEA) was conducted to preliminarily investigate the role of immune cells in the disease process of OA. In this study, we used the 'GSVA' package to conduct ssGSEA analysis to clarify the immune landscape in OA cartilage tissues and the relationship between signature gene expression and the level of immune cell infiltration. Twenty-eight types of immune cells and their characteristic markers were identified from the results of previous studies (*Subramanian et al., 2005*). Beyond the initial analysis with the GSE114007 dataset, we also conducted immune cell infiltration and GRPEL1 correlation analysis on the validation dataset GSE57218.

## Cartilage primary cell culture and cell models of OA

The isolation of mouse chondrocytes was performed according to previously published methods (*Jin et al., 2021*). Specifically, 10 four-week-old healthy and normal C57BL/6J mice were obtained from Zhengzhou University and housed in the SPF animal house. The mice were euthanized at the time of the experiment using an overdose of sodium pentobarbital. Samples of articular cartilage from the knee and femoral heads of mice were collected, cut into pieces, and washed with sterile PBS. This was followed by digestion using trypsin for 30 min, removal of trypsin, and washing with PBS. The samples were then digested using type II collagenase for 5 h, filtered, and the liquid retained. Finally, culture medium was added for incubation. Chondrocytes were cultured in DMEM/F12 medium (10091; Procell, Wuhan, China) containing 10% FBS (10091; Gibco, Billings, MT, USA) and maintained at 37 °C with 5% $CO_2$. To simulate osteoarthritis *in vitro*, the second-generation chondrocytes were treated with 5 μg/mL LPS (L2880-10MG; Sigma-Aldrich, St. Louis, MO, USA) based on literature references (*Chen et al., 2020*; *Li &*

*Chen, 2021*). The cells were treated with LPS for 24 h as the experimental group, while the untreated cells were used as the control group. The success of the model was confirmed by RT-qPCR, and subsequent experiments were conducted using third-generation cells. The guidelines for the care and use of laboratory animals are conducted in all procedures involving animals. Measures to minimize animal suffering included the use of anesthesia during chondrocyte isolation and providing appropriate post-operative care. And the animal experiment protocol was approved by the Zhengzhou Weisha Biotechnology Co. Laboratory Animal Ethics Committee (Approval number: V3A02022109012).

## Quantitative Real-time PCR
Total RNA was extracted from cells using the Total RNA Kit I (R6834-02; Omega, Norcross, GA, USA), and its quality and purity were assessed *via* NanoDrop (Thermo Fisher Scientific, Waltham, MA, USA). The extracted RNA was then reverse-transcribed into cDNA using the NovoScript Plus cDNA SuperMix kit (E047-01B; Novoprotein, Beijing, China). The expression of the signature gene was quantified using the Novostart SYBR qPCR SuperMix (E096-01A; Novoprotein), and the $2^{-\Delta\Delta CT}$ method was employed for result calculation. The primer sequence can be found in Table S2.

## Cell Transfection
One day before transfection, OA cells were seeded in 6-well plates to reach 70–80% confluency at the time of transfection. Transfection was performed using Lipofectamine 3000 (L3000015; Invitrogen) according to the manufacturer's instructions. The experiment was divided into two groups: the control group (NC group, pCMV-EGFP- 3×FLAG-Neo vector) and the overexpression group (OE group, pCMV-EGFP-GRPEL1- 3×FLAG-Neo vector). For each well, 2.5 μg of the respective plasmid and 5 μl of Lipofectamine 3000 were mixed and added. After 6 h of transfection, the medium was replaced with complete medium containing 10% FBS, and the cells were cultured for an additional 48 h. Transfection efficiency was verified by RT-PCR to measure GRPEL1 expression levels.

## Statistical methods
The statistical analyses conducted in this study utilized the R language (version 4.1.1; *R Core Team, 2021*). A comparison between the two groups was made using either the student's *t*-test or the Wilcoxon test. Spearman's correlation was employed to assess the relationship between the two variables. A significance level of $p < 0.05$ was deemed indicative of statistical significance.

# RESULTS
## Results of differentially expressed genes
To explore the genes that play potentially key roles in the pathological process of OA, the data microarray set GSE114007 was first subjected to differential expression analysis, and the results revealed a vast number of genes with abnormal expression, with a total of 1,502 DEGs in the OA cartilage samples. Among them, there were 790 genes with increased expression and 712 genes with decreased expression in OA, as shown in Fig. 1 and Table S1.

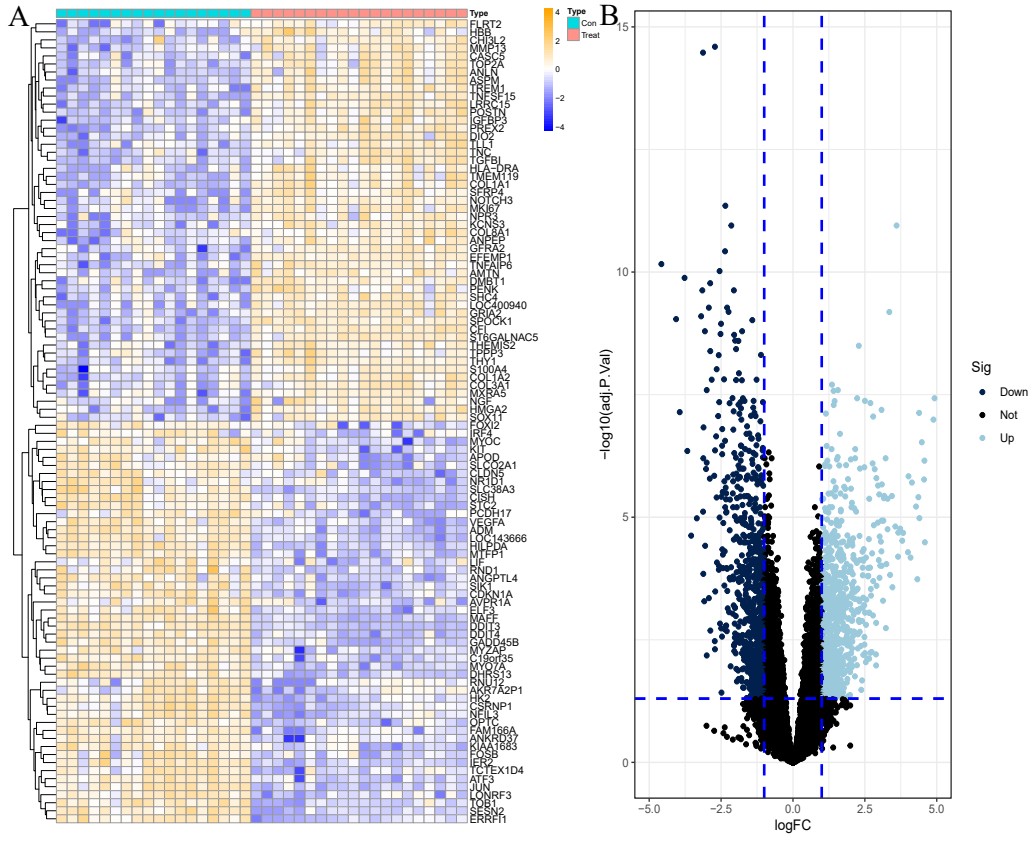

**Figure 1** **Differentially expressed genes (DEGs) in OA cartilage samples.** (A) Heatmap of DEGs. Orange represents up-regulated genes, blue represents down-regulated genes. (B) Volcano diagram of DEGs, light blue represents up-regulated genes, dark blue represents down-regulated genes.

## Functional enrichment analysis of differentially expressed genes

We conducted a GO analysis to annotate their functions based on the above-screened DEGs. In this analysis, we included all differentially expressed genes, both upregulated and downregulated, to preliminarily explore their biological functions in the pathological process of OA. Among them, biological process was mainly enriched in extracellular matrix organisation, extracellular structure organisation, external encapsulating strcture organisation, ossification, *etc*. Cellular component is mainly enriched in collagen-containing extracellular matrix, basement membrane, collagen trimer, endoplasmic reticulum lumen, transcription regulator complex, *etc*. Molecular function is mainly enriched in extracellular matrix structural constituent, conferring tensile strength glycosaminoglycan binding, heparin binding, DNA-binding transcription activator activity, RNA-polymerase II-specific, sulfur compound binding, and so on (Figs. 2A and 2B). These enriched processes are closely related to the pathological progression of OA. The degradation and remodeling of the extracellular matrix are hallmark features of OA, while abnormal ossification accelerates cartilage degeneration (*Rahmati et al., 2017*; *Fujii et al., 2022*). Additionally, the degradation of collagen and glycosaminoglycans

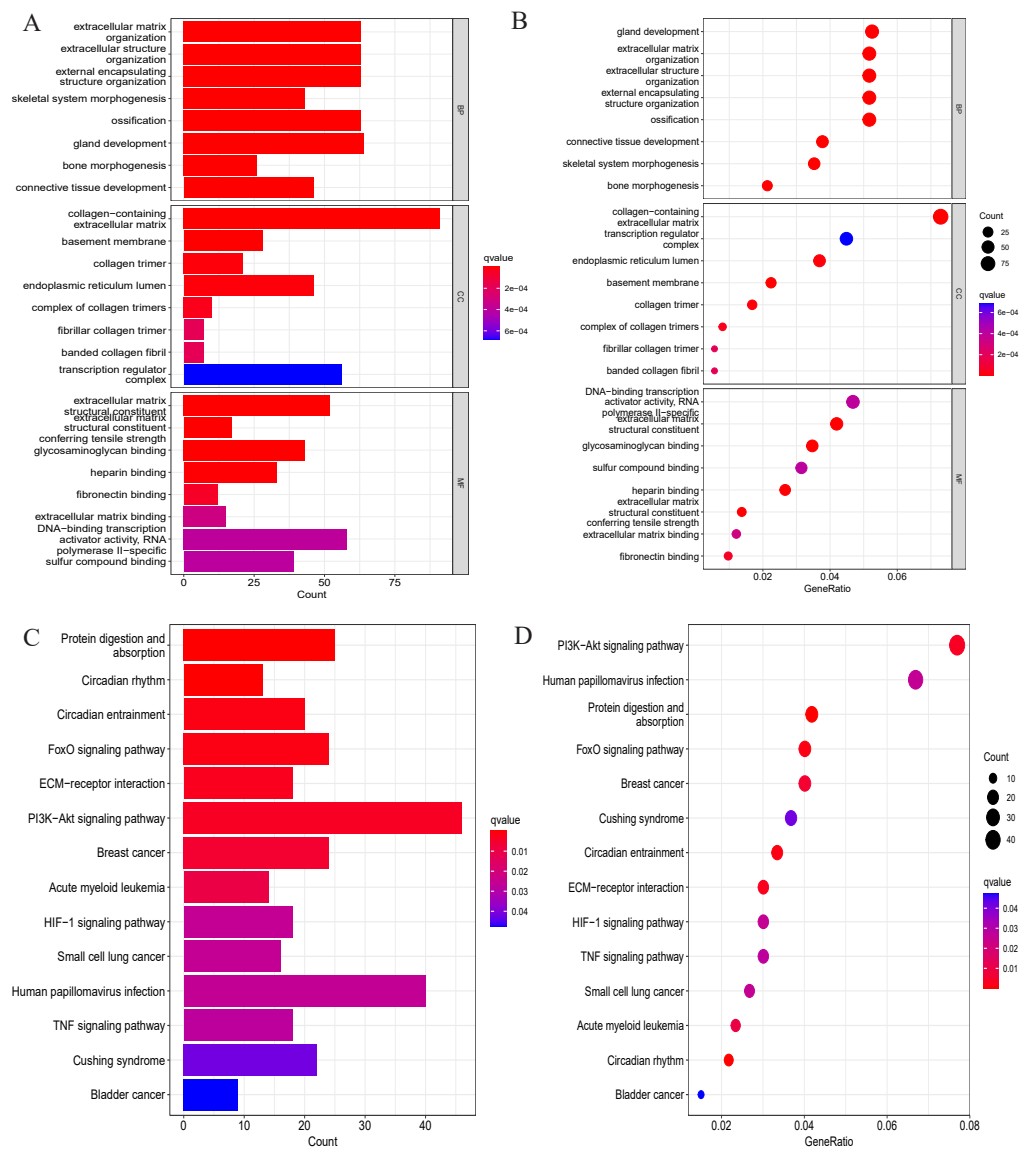

**Figure 2   GO and KEGG enrichment analysis results of DEGs.** (A) Bar chart for GO analysis results; (B) bubble chart for GO analysis results; (C) bar chart for KEGG analysis results; (D) bubble chart for KEGG analysis results.

significantly disrupts the articular cartilage's structural integrity and mechanical properties. This degradation process leads to the loss of cartilage matrix and functional impairment, resulting in joint instability, uneven load distribution, and cartilage erosion, ultimately driving the pathological progression of OA (*Charlier et al., 2019*; *Chen et al., 2023*). These findings provide crucial biological insights into the pathological mechanisms of OA.

After that, to explore the pathways through which DEGs were involved in the pathological process of OA, KEGG analysis was also conducted in this study, and the results are shown in Figs. 2C and 2D. Specifically, they were mainly enriched in signaling pathways such

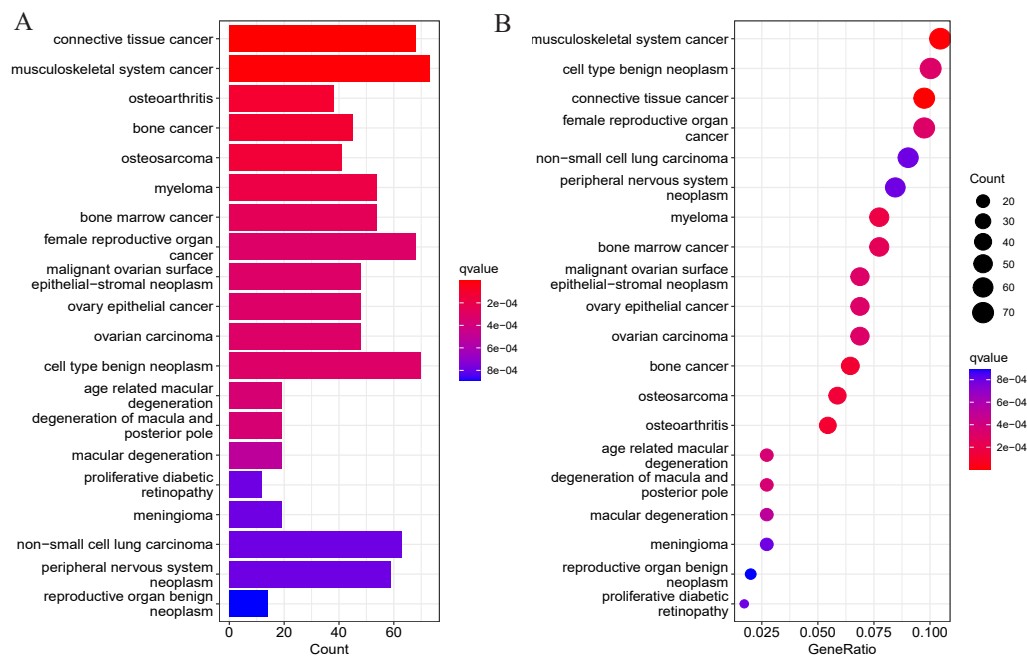

**Figure 3** **Results of DO enrichment analysis of DEGs.** (A) Bar chart of the analyzed results; (B) bubble diagram of the analyzed results.

as the FoxO signaling pathway, ECM-receptor interaction, PI3K-Akt signaling pathway, HIF-1 signaling pathway, and TNF signaling pathway.

## Disease ontology analysis of differentially expressed genes

To understand whether DEGs may function in the pathological process of OA, DO analysis was performed on the DEGs screened above. The results suggested DEGs were mainly enriched in diseases such as osteoarthritis, tumors of the musculoskeletal system, and malignant tumors of other systems (Fig. 3). This also suggests the presence of potential signature genes of OA in these DEGs.

## LASSO regression analysis and SVM-RFE algorithm to select OA signature genes

The aforementioned findings suggest the presence of certain individuals within the differentially expressed genes (DEGs) who possess crucial roles in the initiation and advancement of OA. Consequently, this study aims to conduct comprehensive analyses to identify the signature genes associated with OA. Firstly, the results of differential expression analysis revealed 1502 DEGs in the cartilage tissues of OA patients. Secondly, to further narrow down the range of feature genes, the intersection of DEGs and MRGs was taken, and 45 potential signature genes for the disease were finally obtained (Fig. 4A). And the specific list of signature genes is shown in Table S3. Additionally, enrichment analysis of MRGs within the DEGs revealed significant involvement in mitochondrial transmembrane transport, regulation of mitochondrial organization, and oxidoreductase activity. These findings suggest that mitochondrial dysfunction plays a pivotal role in the progression of
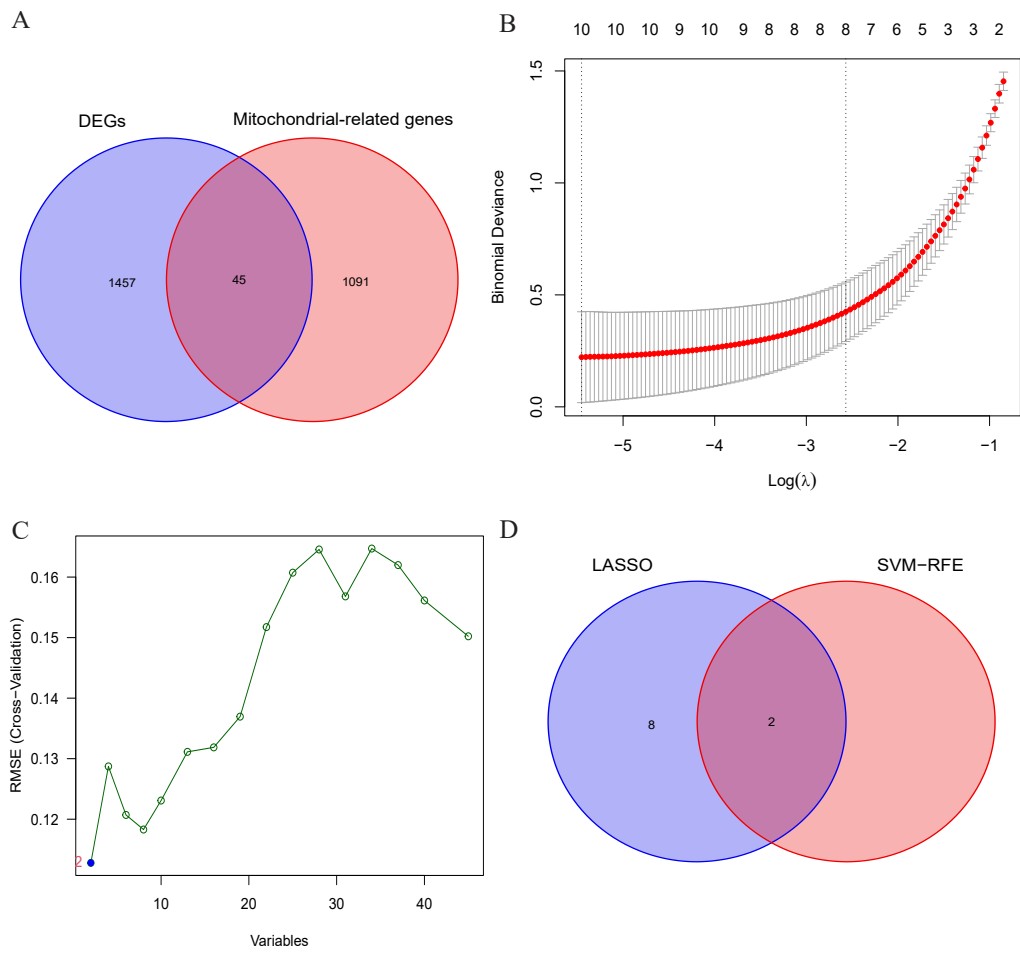

**Figure 4** **Screening of OA signature genes.** (A) Wayne diagram for DEGs and mitochondria-related genes (MRGs) to take the intersection and obtain differentially expressed mitochondria-related genes (DMRGs); (B) LASSO model screening for OA signature genes in DMRGs; (C) SVM-RFE screening for OA signature genes in DMRGs; (D) Wayne's plot takes the intersection of the signature genes screened by the two methods.

OA, providing potential targets for therapeutic intervention (Fig. S1). Subsequently, 10 and 2 potential signature genes were determined respectively by using LASSO analysis and SVM-RFE algorithms (Figs. 4B, 4C). Finally, after cross-referencing the above results, two genes (MTFP1, GRPEL1) were identified. Therefore, these 2 genes were initially identified as signature genes of OA (Fig. 4D).

## Differential expression analysis of signature genes and diagnostic efficacy for diseases

To specify the expression levels of OA signature genes in cartilage tissues, differential expression analysis was conducted. Firstly, the expression of MTFP1 and GRPEL1 were extracted from the experimental group dataset GSE114007 for differential analysis. The results revealed the expression of MTFP1 and GRPEL1 were abnormally downregulated in the OA group (Figs. 5A and 5B). However, it is not known whether they are specific

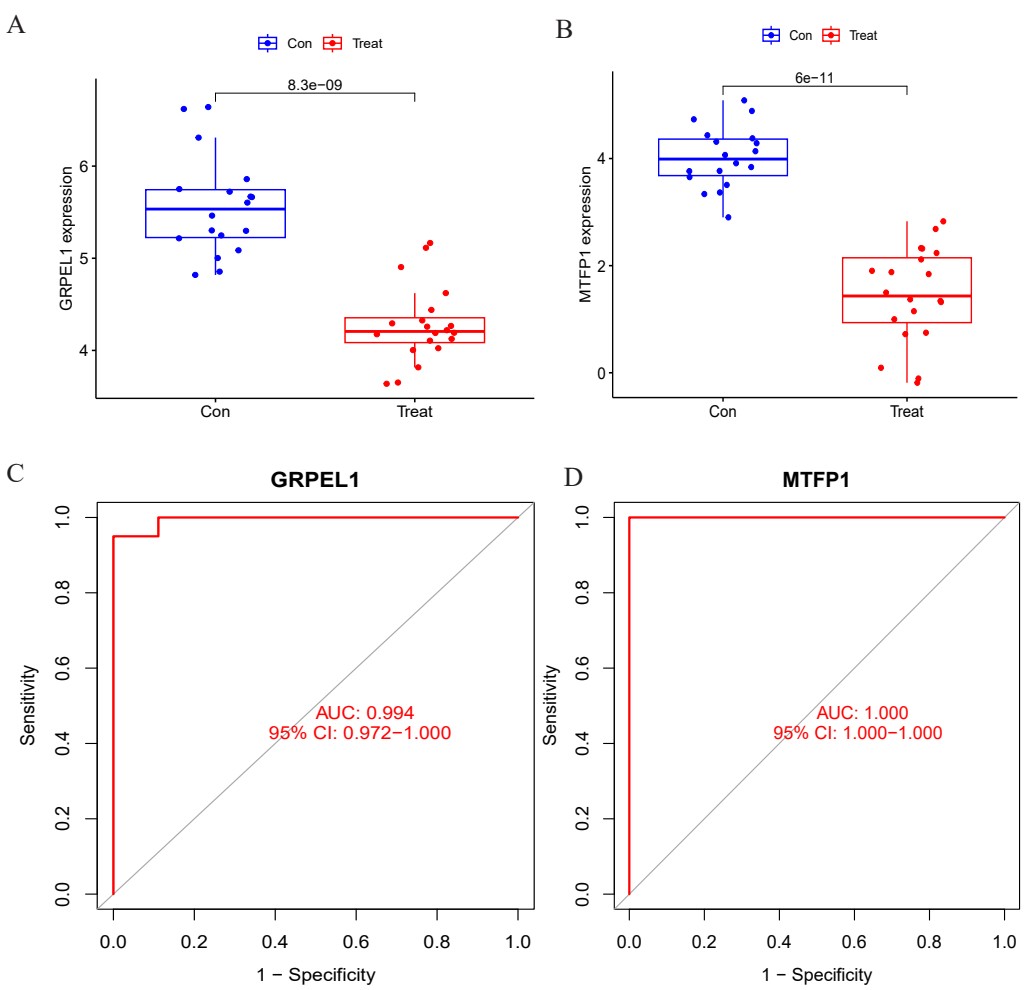

**Figure 5** **Differential expression analysis of signature genes and diagnostic efficacy for diseases.** (A–B) Expression levels of the two signature genes in the experimental group dataset. Note: Con: healthy control samples; Treat: OA samples. (C–D) ROC curves analyzing the diagnostic efficacy of the two signature genes for OA in the experimental dataset.

and whether they have diagnostic value for OA. Therefore, ROC curves were plotted to clarify this issue. However, it is not known whether they are specific and whether they have diagnostic value for OA. Therefore, ROC curves were plotted to clarify this issue. The results showed AUC of GRPEL1 was 0.994, and that of MTFP1 was 1 (Figs. 5C, 5D). This implies that GRPEL1 and MTFP1 have good diagnostic values for OA. In conclusion, the above results suggest GRPEL1 and MTFP1 can be used as signature genes of OA and have better diagnostic efficacy.

## Validation of signature gene expression and diagnostic efficacy

To clarify the accuracy of the above findings, the expression of MTFP1 and GRPEL1 in the test group dataset GSE57218 was extracted for differential analysis. The results showed only GRPEL1 expression was abnormally down-regulated in the OA group (see Figs. 6A, 6B).

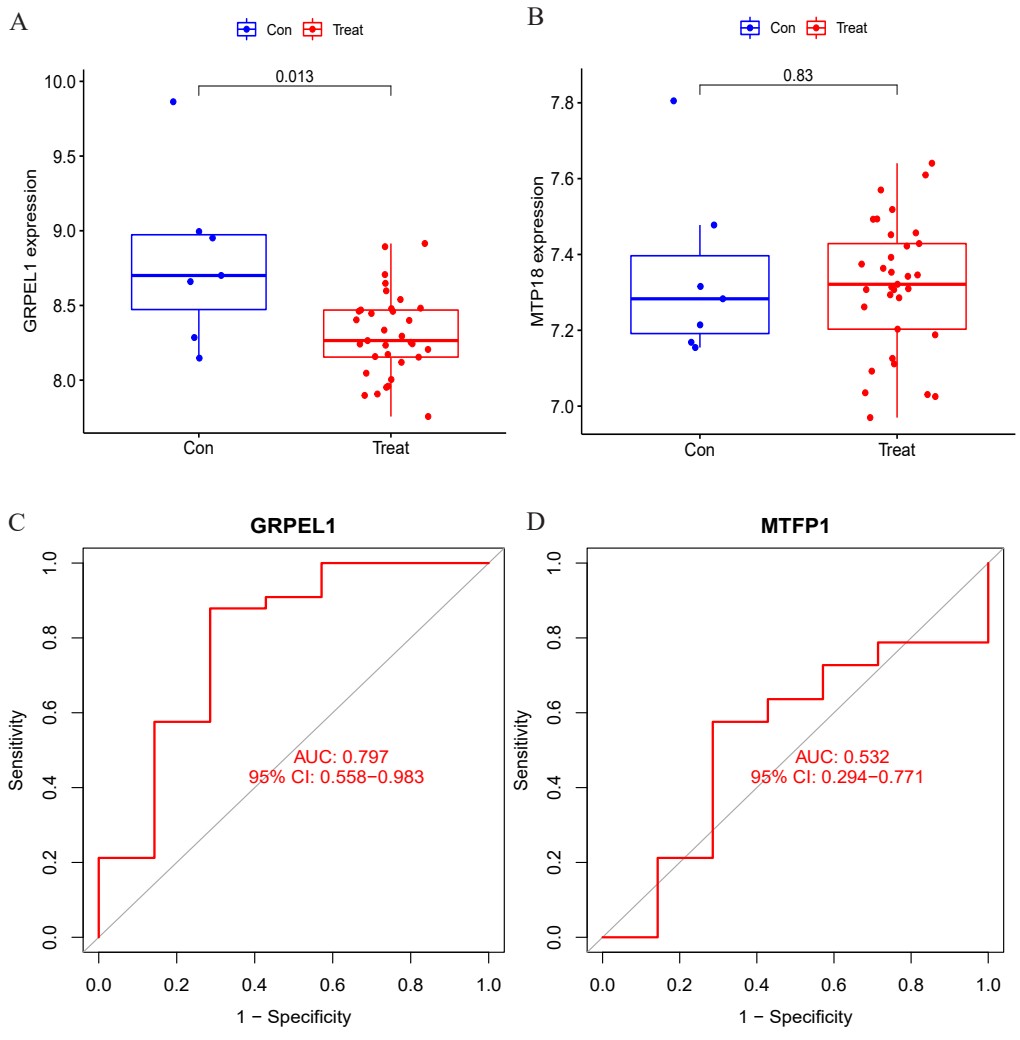

**Figure 6 Validation of the expression and diagnostic efficacy of the signature genes.** (A–B) Expression levels of the two signature genes in the validation group dataset; (C–D) ROC curves analyzing the diagnostic efficacy of the two signature genes for OA in the experimental group dataset.

Similarly, the diagnostic efficacy of GRPEL1 for OA was further verified by ROC curves, and the AUC value was determined to be 0.797 (Figs. 6C, 6D). This indicated that GRPEL1 also had some diagnostic value for OA in the test group.

Further, to further verify the reliability of the results, this study used mouse chondrocytes to construct an *in vitro* OA model to verify the expression of the signature genes. First, the changes of extracellular matrix and degradative enzymes in chondrocytes after 5 ug/ml LPS treatment were detected by RT-qPCR, and the results revealed the expression of Ifngr2, Adamts5 and Mmp13 increased, and the expression of aggrecan and collagen II decreased, suggesting the model was successfully constructed, as shown in Fig. 7A. Subsequently, RT-qPCR was performed to detect the expression of Grpel1, and the results indicated its expression was reduced in the experimental group, as shown in Fig. 7B.

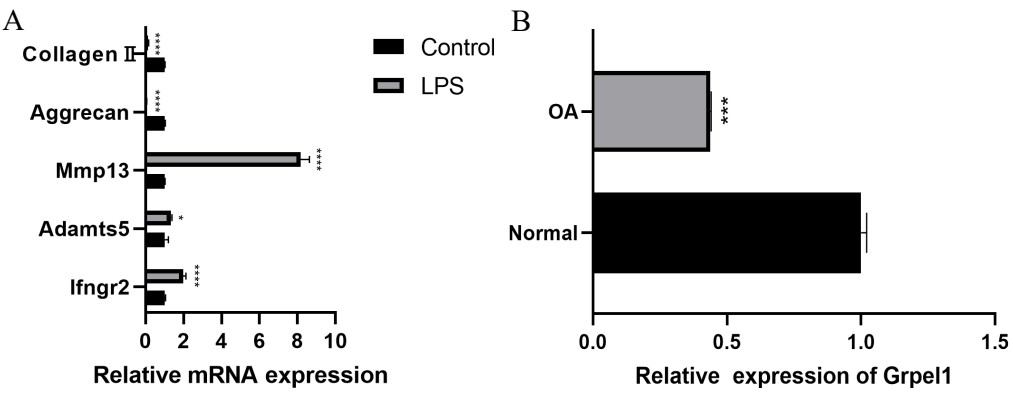

**Figure 7** Validation of Grpel1 expression in an *in vitro* model of OA. (A) RT-qPCR to verify the success of *in vitro* model establishment; (B) RT-qPCR showed Grpel1 expression was reduced in the *in vitro* OA model.

The above results suggest that GRPEL1 has a reliable diagnostic efficacy for OA. Specifically, GRPEL1 expression level was found to be a useful diagnostic marker based on our analysis.

## Impact of Grpel1 overexpression on chondrocyte function

To further investigate the potential role of Grpel1, primary OA cells were transfected to overexpress Grpel1, and the expression changes of representative genes, including extracellular matrix components, matrix-degrading enzymes, and inflammatory molecules, were detected using RT-qPCR. These genes included Mmp13, Adamts5, Col2a1, Ifngr2, and Aggrecan. The results are shown in Fig. 8. Compared to the control group, Grpel1 overexpression significantly upregulated the expression of Col2a1 and Aggrecan, while significantly downregulating the expression of Mmp13, Adamts5, and Ifngr2. These findings suggest that Grpel1 may play a protective role in the pathogenesis of OA by promoting cartilage matrix synthesis, inhibiting matrix degradation, and reducing inflammatory responses.

## Relationship between signature gene expression and level of immune cell infiltration

Immune cells are important in the prevention and treatment of OA through the secretion of cytokines and chemokines that promote inflammatory responses and articular cartilage destruction (*Chen et al., 2021a*). To investigate the degree of immune cell infiltration in OA, ssGSEA analysis was conducted on the experimental group data set. First, a difference analysis was undertaken on the level of immune cell infiltration in the OA and control groups, as shown in Fig. 9A. The results revealed the infiltration levels of activated B cells, eosinophils, type 17 T helper cells, memory B cells, central memory CD8 T cells, and effector memory CD8 T cells were decreased in OA samples compared to normal samples. In contrast, infiltration levels of activated dendritic cells, gamma-delta T cells, immature dendritic cells, MDSCs, macrophages, natural killer T cells, regulatory T cells, T follicular

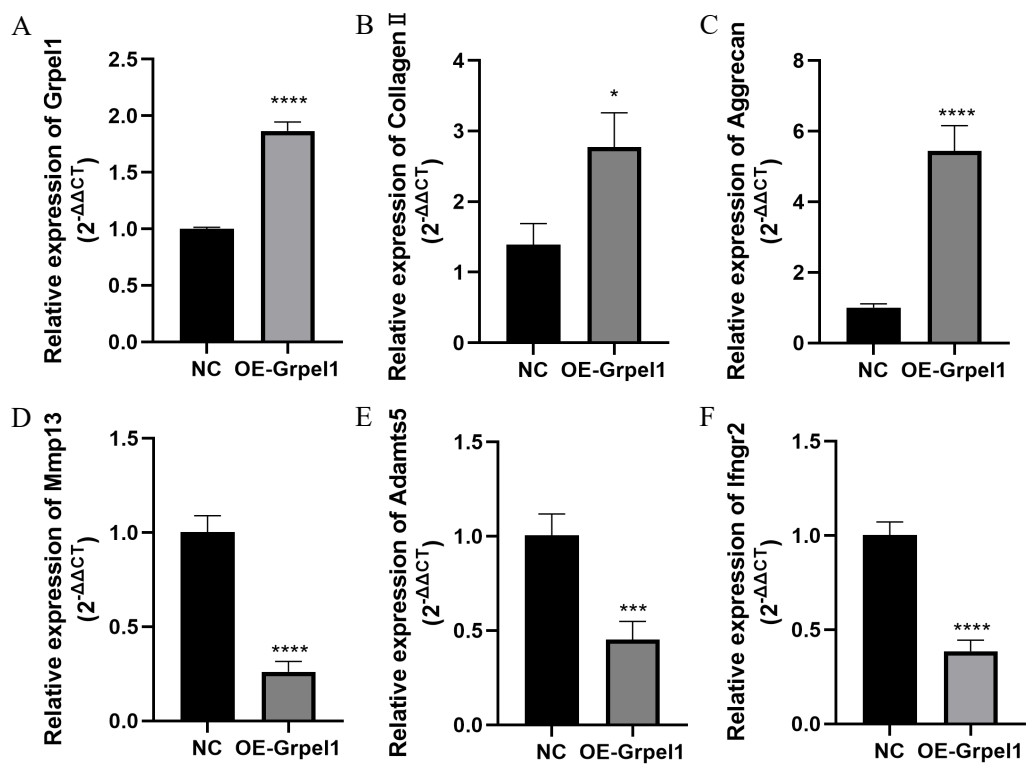

**Figure 8** **Impact of Grpel1 overexpression on chondrocyte function.** Impact of Grpel1 overexpression on chondrocyte function. (A) RT-qPCR analysis to verify transfection efficiency; (B–F) RT-qPCR analysis to detect the expression levels of Collagen II, Aggrecan, Mmp13, Adamts5, and Ifngr2 after Grpel1 overexpression.

helper cells, and type 2 T helper cells were increased in OA samples. Summarily, multiple immune cells undergo infiltration abundance alterations in the OA immune landscape. To initially clarify whether the involvement of GRPEL1 in the pathological process of OA is related to immune cells, further correlation analyses were undertaken. The findings of this study demonstrate a positive correlation between GRPEL1 and the infiltration level of certain immune cells, namely MDC and macrophages, while a negative correlation was observed with the infiltration level of NK cells, as depicted in Fig. 9B.

Additionally, immune cell infiltration and correlation analyses were conducted based on the validation dataset GSE57218, as shown in Fig. S2. Specifically, the infiltration abundance of MDSCs, macrophages, and T follicular helper cells was higher in OA tissues, consistent with the results from the experimental dataset. However, the correlation analysis between GRPEL1 and immune cell infiltration did not show consistency with the results from the experimental dataset.

## DISCUSSION

OA is a prevalent condition that involves the degeneration of joints over time, which in severe cases can lead to varying degrees of disability (*Woodell-May & Sommerfeld,*

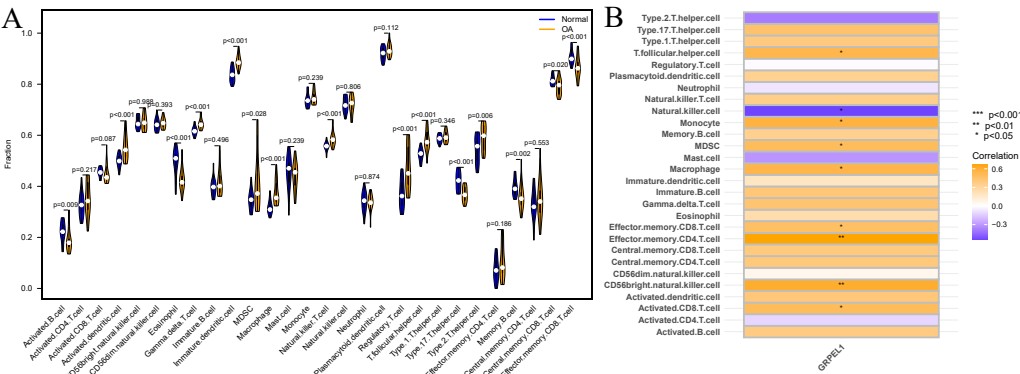

**Figure 9  Immune cell infiltration analysis based on the experimental group dataset GSE114007.** (A) Differences in immune cell infiltration of 28 immune cells between OA and normal samples; (B) correlation between GRPEL1 and abundance of immune cell infiltration.

*2020*). Treatment of OA is mainly based on oral medications and end-stage arthroplasty. However, there is currently no effective method of reversing OA other than late-stage surgery. Therefore, early diagnosis and early intervention would greatly improve or restore patients' joint function, delay disease progression and avoid surgical treatment. Recently, bioinformatics has played an increasingly critical role in the identification of OA biomarkers. For instance, SLC3A2, a ferroptosis-related gene, was identified by WGCNA as a valuable OA biomarker with reduced expression in chondrocytes, which promotes the onset of chondrocyte ferroptosis and degeneration (*Liu et al., 2022a*). *Liu et al. (2022b)* constructed an OA diagnostic model consisting of six macrophage-associated genes using machine learning, which has good diagnostic efficacy and may be a potential immunotherapeutic target. However, it is not known whether there are signature genes in the MRGs that are of high value for early diagnosis and intervention of OA.

Therefore, a series of comprehensive analyses were undertaken to identify the signature genes underlying OA in MRGs. First, 1,502 genes were found to be abnormally expressed by differential analysis. Further functional enrichment analysis revealed that these DEGs may be involved in the pathological process of OA mainly by regulating the PI3K-Akt signaling pathway and the extracellular matrix (ECM). This suggests that these DEGs may contain signature genes of OA since the above signaling pathways have an irreplaceable impact on the pathological mechanisms of OA, which has been substantiated by previous research. For example, the PI3K-Akt signaling pathway participates in the normal metabolism of joint tissues but also contributes to the progression of OA by regulating biological processes such as chondrocyte proliferation, apoptosis, autophagy, and inflammatory responses (*Sun et al., 2020*). This homeostasis of articular cartilage ECM synthesis and bone remodeling is essential for chondrocyte and joint health (*Rahmati et al., 2017*). Notably, the mitochondrial respiratory chain is the main energy supplier in this homeostasis during this process, and its deficiency is the main cause of impaired cartilage ECM integrity and mechanical stability *in vivo* (*Rahmati et al., 2017*). Therefore, mitochondrial respiratory chain deficiency is the major cause of impaired cartilage ECM integrity and mechanical

stability *in vivo* (*Rahmati et al., 2017*). Taken together, it is reasonable to believe there may be signature genes for OA among these DEGs that deserve further investigation.

Next, the combined LASSO and SVM-RFE algorithms successfully identified two signature genes with potential value. Afterward, the expression levels and diagnostic efficacy of the two signature genes were assessed by difference analysis and ROC curves using transcriptomic data from the experimental and test groups. Encouragingly, GRPEL1 exhibited stable low expression levels in both datasets and had good diagnostic value for OA, and thus was identified as a signature gene for OA. Previous studies have shown that GRPEL1 overexpression promotes the binding of GRPEL1 to mtHSP70, maintaining mitochondrial protein homeostasis and function (*Bubb et al., 2021*). Given that mitochondrial dysfunction and oxidative stress are known contributors to cartilage degeneration in OA, and that mitochondrial dysfunction exacerbates oxidative stress, we speculate that GRPEL1 may slow OA progression by maintaining mitochondrial function and protecting chondrocytes from stress-induced damage and inflammation. Surprisingly, the role of GRPEL1 in OA has not yet been reported, suggesting that this gene not only has high diagnostic value but also provides new directions for future research. To validate these findings, we examined the expression of Grpel1 in an *in vitro* model of OA using RT-qPCR and further investigated the effects of Grpel1 overexpression on several key genes, including cartilage matrix synthesis markers (Col2a1 and Aggrecan), matrix-degrading enzymes (Mmp13 and Adamts5), and an inflammatory marker (Ifngr2). These preliminary results provide important experimental evidence supporting Grpel1 as a potential therapeutic target for OA.

Given that immune cells play an imperative role in the genesis and progression of OA (*Weber, Chan & Wen, 2019*), we explored the infiltration of different immune cells in OA and their relationship with GRPEL1 expression using the GSE114007 dataset. The results of ssGSEA analysis demonstrated a significant increase in activated dendritic cells, gamma delta T cells, immature dendritic cells, MDSCs, macrophages, and natural killer T cells in OA samples compared to normal samples. Additionally, regulatory T cells, T follicular helper cells, type 2 T helper cells, and other immune cells were significantly increased. The role of these immune cells in the development of OA has been confirmed by previous studies. For example, an increased abundance of NK cell infiltration leads to the development of synovial inflammation, which is involved in the progression of OA (*Crinier et al., 2020*). Macrophages differentiate into either the M1 type, which promotes inflammation, or the M2 type, which inhibits inflammation after the onset of OA (*Sun et al., 2020*). Targeting macrophage polarization to inhibit synovial M1 macrophage activity in OA joints is expected to be key to overcoming OA. Additionally, immune cells such as T follicular helper cells and type 2 T helper cells may also be involved in OA pathological processes by releasing cytokines (*Shan et al., 2017*; *Fernandes et al., 2020*).

Correlation analysis indicates high GRPEL1 expression is significantly associated with increased MDSCs and macrophage infiltration levels. This may be due to GRPEL1 influencing the communication between chondrocytes and these cells, thereby regulating their proliferation and migration. Additionally, NK cells regulate immune responses by directly killing diseased cells and secreting cytokines (*Vivier et al., 2024*). A reduction

in NK cells may lead to weakened immune surveillance of synovial inflammatory cells and chondrocytes, thus exacerbating inflammation and tissue destruction (*Zheng et al., 2024*). This reduction may be related to GRPEL1's regulation of immunosuppressive signaling pathways, thereby playing a role in OA progression. While the decrease in NK cells and the increase in MDSCs and macrophages may seem to represent differing aspects of the immune response, they reflect the complexity of the OA immune microenvironment. GRPEL1 inhibits the activity and infiltration of NK cells, creating an immunosuppressive environment favorable for MDSCs and macrophages. This enhances their immunoregulatory functions and further exacerbates inflammation and cartilage degradation. However, these hypotheses require further evidence for validation.

In comparing the GSE114007 and GSE57218 datasets, we found that although the ssGSEA analysis results for MDSCs, macrophages, and T follicular helper cells showed a high degree of consistency, the correlation between GRPEL1 and immune cell infiltration did not exhibit the same consistency. This discrepancy may be due to several reasons. First, differences in sample sources, processing methods, and sequencing technologies between datasets may lead to variations in the measurement of gene expression and immune cell infiltration levels. Second, osteoarthritis is a highly heterogeneous disease, and the pathological processes and immune responses can vary significantly among different patients. Additionally, the immune microenvironment of OA is very complex, involving interactions among various immune cells, cytokines, and signaling pathways, which may result in dynamic changes in immune cell infiltration and gene expression across different patients and tissue samples. In summary, GRPEL1 modulates the activity of immune cells through different mechanisms, playing an important role in the progression of OA. It offers potential as an immunotherapeutic target for OA, providing a theoretical basis for OA immunotherapy.

In summary, this study identified a novel diagnostic marker for OA through the joint comprehensive analysis of multiple datasets and multiple analytical methods. Although this study is the first to identify GRPEL1 as a signature gene diagnostic marker for OA, which is innovative and potentially applicable, there are some limitations. Firstly, the heterogeneity of data sources is unavoidable, as this study used public data from multiple centers. Second, despite providing directions for future research, this study failed to explore in depth how GRPEL1 affects the occurrence and development of OA. Although the sample size of the validation dataset (GSE57218) is relatively small (seven normal samples and 33 OA samples), it is reasonable and valuable in the context of osteoarthritis research. We used robust analytical methods to minimize the impact of the small sample size and conducted cross-validation to enhance reliability. Nevertheless, to further confirm our results and enhance their generalizability, we recommend future studies to validate our findings using larger, independent cohorts. Finally, as this study was retrospective and lacked new clinical samples and data, subsequent large-sample, multi-center clinical trials are needed to further confirm the specificity and reliability of the findings.

In summary, this study identified a novel diagnostic marker for OA through the joint comprehensive analysis of multiple datasets and multiple analytical methods. Although this study is the first to identify GRPEL1 as a signature gene diagnostic marker for OA,

which is innovative and potentially applicable, there are some limitations. Firstly, the heterogeneity of data sources is unavoidable, as this study used public data from multiple centers. Second, despite providing directions for future research, this study failed to explore in depth how GRPEL1 affects the occurrence and development of OA. Although the sample size of the validation dataset (GSE57218) is relatively small (seven normal samples and 33 OA samples), it is reasonable and valuable in the context of osteoarthritis research. We used robust analytical methods to minimize the impact of the small sample size and conducted cross-validation to enhance reliability. Nevertheless, to further confirm our results and enhance their generalizability, we recommend future studies to validate our findings using larger, independent cohorts. Finally, as this study was retrospective and lacked new clinical samples and data, subsequent large-sample, multi-center clinical trials are needed to further confirm the specificity and reliability of the findings.

## CONCLUSIONS

The current study elucidates GRPEL1 as a potential diagnostic biomarker for OA and provides insights into therapeutic targets for future research. These findings not only contribute to a deeper understanding of the pathophysiology of OA, but also provide theoretical foundations and practical implications for the development of novel diagnostic and therapeutic strategies.

**Abbreviations**

| | |
|---|---|
| **OA** | Osteoarthritis |
| **MRGs** | Mitochondria-related genes |
| **DEGs** | Differentially expressed genes |
| **MDEGs** | Mitochondria DEGs |
| **GEO** | Gene Expression Omnibus |
| **GO** | Gene Ontology |
| **KEGG** | Kyoto Encyclopedia of Genes and Genomes |
| **DO** | Disease Ontology |
| **LASSO** | Least absolute shrinkage and selection operator |
| **SVM-RFE** | Support vector machine-recursive feature elimination |
| **ROC** | Receiver operating characteristic |
| **AUC** | Area under the curve |
| **ssGSEA** | Single-sample gene set enrichment analysis |
| **ECM** | Extracellular matrix |

### Funding

This study was supported by the National Natural Science Foundation of China (No. 82172438). The funders had no role in study design, data collection and analysis, decision to publish, or preparation of the manuscript.

## Grant Disclosures
The following grant information was disclosed by the authors:
The National Natural Science Foundation of China: No. 82172438.

## Competing Interests
The authors declare there are no competing interests.

## Author Contributions
- Hongbo Wang conceived and designed the experiments, performed the experiments, prepared figures and/or tables, and approved the final draft.
- Zongye Zhang performed the experiments, analyzed the data, authored or reviewed drafts of the article, and approved the final draft.
- Xingbo Cheng performed the experiments, analyzed the data, prepared figures and/or tables, and approved the final draft.
- Zhenxing Hou performed the experiments, prepared figures and/or tables, and approved the final draft.
- Yubo Wang conceived and designed the experiments, performed the experiments, prepared figures and/or tables, and approved the final draft.
- Zhendong Liu conceived and designed the experiments, authored or reviewed drafts of the article, and approved the final draft.
- Yanzheng Gao conceived and designed the experiments, authored or reviewed drafts of the article, and approved the final draft.

## Animal Ethics
The following information was supplied relating to ethical approvals (i.e., approving body and any reference numbers):

The animal experiment protocol was approved by the Zhengzhou Weisha Biotechnology Co. Laboratory Animal Ethics Committee (Approval number: V3A02022109012).

## Data Availability
The data is available at NCBI GEO: GSE114007, GSE57218, and Zenodo: Hongbo Wang. (2024). Machine learning algorithm-based biomarker exploration and validation of mitochondria-related diagnostic genes in osteoarthritis. https://doi.org/10.5281/zenodo.12104002.

## Supplemental Information
Supplemental information for this article can be found online at http://dx.doi.org/10.7717/peerj.17963#supplemental-information.

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
