# Peer review of "Machine learning algorithm-based biomarker exploration and validation of mitochondria-related diagnostic genes in osteoarthritis"

_PeerJ, doi:10.7717/peerj.17963_

## Round 0.1 · original submission · Major Revisions

This manuscript is being returned to you for major revisions. Please address all the concerns raised by reviewers. For resubmission, provide a detailed point-by-point response to the concerns raised by reviewers and highlight changes to the manuscript. Please ensure relevant literature is thoroughly covered and cited. Provide a detailed explanation for how your work adds value to the field in light of other similar published studies.

Reviewer 1 ·

Basic reporting

This study aimed to identify diagnostic signature genes associated with osteoarthritis (OA) from a set of mitochondria-related genes (MRGs). Gene expression profiles of OA cartilage were analyzed using different methods, including differential gene expression analysis, Gene Ontology (GO) and Kyoto Encyclopedia of Genes and Genomes (KEGG) enrichment analysis. The results identified GRPEL1 as a potential diagnostic indicator for OA, with low expression levels in OA samples and confirmed through RT-qPCR analysis. Additionally, alterations in immune cell infiltration were observed in OA cartilage tissue, correlating with GRPEL1 expression. These findings provide new insights into potential targeted therapeutic interventions for OA.
The manuscript is well written and the figures are easy to understand.

Experimental design

Three main issues about the analysis method:
1. The threshold used for the DEG analysis in this study, Log2FC <0.5, is relatively low compared to other studies. It is suggested that the authors increase the threshold to 1 or 1.5 to determine if it significantly affects the results.

2. On line 195, the authors mention that the screened DEGs were analyzed by GO. It is important for them to specify whether they used only upregulated DEGs or both upregulated and downregulated DEGs in the GO analysis.

3. In the same paragraph, the authors list the changes found in the GO analysis. It would be beneficial if they could provide more biological insights to explain the relevance of the highlighted biological processes to OA.

Validity of the findings

They analyzed two independent dataset and use experiment method to confirm the finding. The conclusion is overall solid.
To validate the findings from GSE114007, the authors should repeat the immune cell infiltration and GRPEL1 correlation analysis in the validation dataset GSE57218.

Additional comments

1. The authors should include a more detailed description of previous findings regarding the biological functions of Grpel1 and propose a mechanism to explain why Grpel1 is related to OA.

2. In line 236, it is stated that the AUC of GRPEL1 was 0.797, but Figure 5 shows the AUC as 0.994.

Reviewer 2 ·

Basic reporting

General Comments:

The manuscript aims to identify and validate mitochondria-related diagnostic genes for osteoarthritis (OA) using machine learning algorithms. The authors have used various bioinformatics and machine learning methods, including LASSO and SVM-RFE algorithms, to identify signature genes related to OA. Although the study provides valuable insights into the understanding of OA and proposes potential diagnostic markers, several issues need to be addressed.

Experimental design

Major Comments:

Sample Size and Validation:

The sample size, particularly for the validation dataset (GSE57218), is small (7 normal and 33 OA samples). The authors should comment on the limitations this imposes on the generalizability of the study.

Methodological Clarity:

The manuscript should elaborate more on the feature selection methods used, particularly the LASSO and SVM-RFE algorithms. Why were these algorithms chosen? How do they complement each other?

Functional Validation:

The manuscript mentions in vitro validation through RT-qPCR. However, how these genes function in the pathway of OA pathogenesis needs to be elucidated. Simply identifying genes as biomarkers is not sufficient.
Immune Infiltration Analysis:

The immune infiltration analysis is interesting but not thoroughly explained. How does immune infiltration relate to the identified signature genes, particularly GRPEL1?

Data Availability:

The manuscript should include information about data availability, especially the raw data used for machine learning algorithms. This is critical for reproducibility.

Animal Ethics:

The section describing animal experiments should more explicitly mention whether any steps were taken to minimize animal suffering.

Validity of the findings

No comments.

Additional comments

Summary:

This manuscript provides an innovative approach to identify diagnostic markers for OA but requires major revisions in terms of methodological clarity, statistical robustness, and functional validation. Once these issues are addressed, the manuscript could make a significant contribution to the field.

Recommendation:

Major Revision.

Reviewer 3 ·

Basic reporting

The author conducted a machine learning pipeline to identify mitochondria-related OA biomarker genes and performed experimental validation in cultured cells. Overall, the manuscript was well-written, and the entire experimental design followed a standard pipeline. The results are well-explained and reasonable.

One of my major concerns is that while the authors want to focus on mitochondria-related genes, the manuscript lacks sufficient information to clarify the importance of MRGs. The enrichment results do not show any typical mitochondrial pathways. If the authors want to explore OA biomarkers, it seems better to apply the ML methods to all genes rather than MRGs only. My suggestion is that the authors should:

1) Review previous studies about the importance of MRGs in OA in the introduction.
2) Add some analyses to show the important role of MRGs in OA with the current datasets, such as any enrichment of MRGs in DEGs.

Experimental design

The author should clarify their strategy for selecting ML methods. Did the author evaluate other methods like decision trees, random forests, and linear/logistic regression?

Although the sample size is small, the author should consider the influence of other clinical factors, such as age, sex, and any other diseases, in all the analyses.

Validity of the findings

No comments

---

## Round 0.2 · accepted · Accept

Please address a minor comment by the reviewer.

One minor point: In the method part, the authors wrote "Screening criteria were Log2FC>=1 and adjust P. Val < 0.05." It should be |Log2FC|>=1.

Reviewer 1 ·

Basic reporting

The authors did a good job in the revision and address all my previous questions. The extra analysis and information makes significantly improve the quality of the paper.
One minor point: In the method part, the authors wrote "Screening criteria were Log2FC>=1 and adjust P. Val < 0.05." It should be |Log2FC|>=1.

Experimental design

no comment

Validity of the findings

no comment

Additional comments

no comment